# Towards Real Robot Learning in the Wild:
# A Case Study in Bipedal Locomotion

**Michael Bloesch\*, Jan Humplik\*, Viorica Patraucean\*, Roland Hafner\*, Tuomas Haarnoja,
Arunkumar Byravan, Noah Siegel, Saran Tunyasuvunakool, Federico Casarini, Nathan Batchelor,
Francesco Romano, Stefano Saliceti, Martin Riedmiller, Ali Eslami, Nicolas Heess**
DeepMind
{bloesch,jhumplik,viorica,rhafner,tuomash,abyravan,siegeln,stunya,fcasarini
batchelor,fraromano,ssaliceti,riedmiller,aeslami,heess}@deepmind.com
*These authors contributed equally

**Abstract:** Algorithms for self-learning systems have made considerable progress in recent years, yet safety concerns and the need for additional instrumentation have so far largely limited learning experiments with real robots to well controlled lab settings. In this paper, we demonstrate how a small bipedal robot can autonomously learn to walk with minimal human intervention and with minimal instrumentation of the environment. We employ data-efficient off-policy deep reinforcement learning to learn to walk end-to-end, directly on hardware, using rewards that are computed exclusively from proprioceptive sensing. To allow the robot to autonomously adapt its behaviour to its environment, we additionally provide the agent with raw RGB camera images as input. By deploying two robots in different geographic locations while sharing data in a distributed learning setup, we achieve higher throughput and greater diversity of the training data. Our learning experiments constitute a step towards the long-term vision of learning "in the wild" for legged robots, and, to our knowledge, represent the first demonstration of learning a deep neural network controller for bipedal locomotion directly on hardware.

**Keywords:** Legged Locomotion, Reinforcement Learning, Vision

## 1 Introduction

Enabling robots to learn and adapt completely autonomously only through interaction with their environment is one of the long-standing goals of AI and robotics. The idea that robots might one day learn and cohabit outside controlled laboratory environments, taking full advantage of the richness of the real world, is appealing especially for mobile robots. However, even though algorithms for self-learning systems have made considerable progress in recent years in terms of data efficiency and robustness (e.g. [1, 2]), concerns around safety and the need for instrumentation (for instance, for reward computation or resets) have so far largely limited experiments with real robots to controlled lab settings [3, 4] or to sim-to-real transfer [5, 6, 7].

While transfer of behaviours from simulation to real can sidestep the challenges around data efficiency and safety, it requires significant engineering effort in the design of high fidelity simulations that represent not only the correct dynamics of the robot but also offer increasingly complex and rich scenarios. More importantly, learning only in simulation exposes the agent to the sim-to-real gap, leaving questions around adaptation to unfamiliar situations after deployment.

Despite first successful applications (e.g. [8]), learning "in the wild" directly on hardware is often thought of as an infeasible method for real legged robots. In this paper, we show that using state-of-the-art model-free deep reinforcement learning (RL) and intrinsically defined rewards, we can learn to walk and avoid walls autonomously in environments with minimal instrumentation. In particular, we show how small humanoid robots can learn to walk and interact with their environment in a learning scenario outside of the laboratory, relying only on on-board sensors and limited prior knowledge of the hardware.

5th Conference on Robot Learning (CoRL 2021), London, UK.

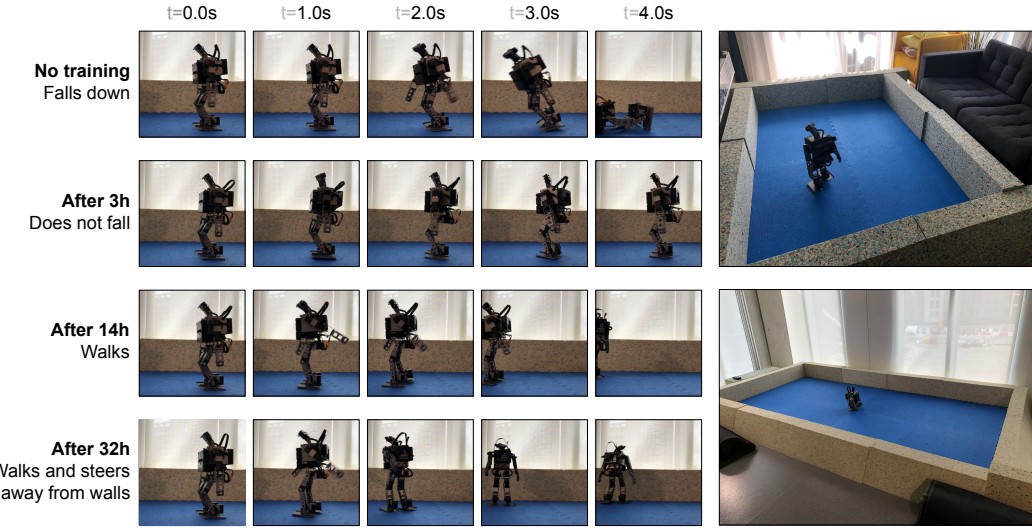

Figure 1: Two robots are trained simultaneously in two different locations depicted on the right. They are sharing the same training data. On the left we show one of the robots and the approximate experiment time it takes for the policy to reach important milestones: maintaining upright position, walking, and eventually walking and avoiding walls. We define experiment time as the total time during which at least on of the robots was on.

The robot uses proprioceptive sensing consisting of an inertial measurement unit (IMU) and joint position encoders together with egocentric RGB vision to perceive its own body and the environment. Rewards and termination signals are derived from the proprioceptive sensors only. We employ a data-efficient off-policy algorithm [1] that allows for distributed training with a variable number of actors. The learning setup does not rely on additional external hardware for state estimation or resets. This allows us to train multiple robots simultaneously, outside of the controlled laboratory environment, in different geographic locations, exposing the robot to a range of environments with varying shape, appearance, and lighting conditions; see Figure 1 (right).

Our results show that such an intrinsic learning signal designed to promote forward movement (combined with the sensory capabilities of the robot and the partly structured environment) is sufficient to learn bipedal gaits and simple wall avoiding behavior requiring only a minimal amount of prior knowledge, see Figure 1 (left) and supplementary video[1]. The distributed training with multiple robots improves data throughput and thus the complexity of the behavior that can be learned. Overall, the results provide a glimpse of the possibility of real-world learning for legged robots.

Our key contributions are: 1. First demonstration of learning a deep non-parametric policy for unconstrained bipedal locomotion on hardware; 2. A data-efficient autonomous learning setup for legged robots that can be easily deployed in different environments and run on multiple robots simultaneously; 3. Learned vision guided legged locomotion with simple obstacle avoidance capabilities.

## 2 Related Work

Conventional approaches to legged locomotion typically rely on models and optimization [9, 10] and are concerned with concepts of gait stability, such as zero moment point [11] and limit cycles [12]. While optimization based methods can yield highly effective solutions, they are laborious to implement, require rigorous domain understanding, and can struggle to generalize to diverse real-world environments and high-dimensional input modalities, such as vision.

Learning and data-driven approaches hold the promise to circumvent these limitation by avoiding the need for accurate modelling. Deep reinforcement learning for locomotion has been studied extensively in simulation [13, 14], and recent papers have shown progress in sim-to-real transfer

---

[1] https://sites.google.com/view/op3-vision-wild

for both quadrupeds [15, 7, 16] and bipeds [17, 6, 5]. Compared to these works, we largely lift the assumption of the availability of a simulation model.

Closest to our approach is the line of work where learning is directly applied on hardware. Unlike ours, these methods consider either a constrained, hand-designed gait parameterization [18, 19, 20, 21] or apply unconstrained deep reinforcement learning to an inherently stable [22] or a quadrupedal robot [3, 4]. None of these works consider an unconstrained humanoid robot or vision observations, and, unlike our approach, they require external instrumentation such as motion capture or other means to quantify the obtained gait.

## 3  Methods

### 3.1  Generalizable Instrumentation and Learning Autonomy

We aim to create a learning setup for a real robot to learn to walk in different *uncontrolled* environments (e.g. households). Consequently, we cannot rely on external sensing such as motion capture systems and we cannot use reset mechanisms such as a gantry. The setup also has to deal with natural, possibly cluttered and irregular environments with variable natural and artificial lighting.

In this paper, we focus on a simpler instantiation of this setting in which the robot is tasked to maintain an upright position, and to move forward as fast as possible in an area surrounded by walls. This forces the robot to use its legs for locomotion while avoiding the walls. Access to camera observations enables the robot to look ahead and adapt its behavior to its surroundings and thereby to deal with a variety of environments. For generality and robustness reasons, we do not employ any (possibly brittle) image pre-processing steps, e.g. for wall detection. Instead, we learn the mapping from the camera pixels to the control signal end-to-end.

The design of our rewards and episode termination conditions is based on [8]. The reward is generated when the feet exhibit a positive forward velocity in the robot's body coordinate frame. To assess this, the velocity of the robot body is computed by assuming that the lowest foot is in contact with the ground and the contact point is stationary (no-slip condition). While these assumptions may not always hold, Hafner et al. [8] have shown experimentally that this reward may still be sufficient to learn forward motion when combined with a reward for keeping the torso upright. This uprightness is in turn estimated from a gravity vector obtained via the IMU's internal orientation estimate. Given that the velocity reward encourages forward movement in the robo-centric coordinate frame, there is a risk of promoting walking in small circles. To avoid this, we penalize large changes in the heading direction, again based on the IMU's orientation estimate. The three reward terms (velocity, uprightness, and no turning) are linearly combined to form the final reward. Episodes are terminated whenever the robot's body inclination exceeds a certain threshold (see Section 4 in the supplementary material for more details about the reward).

Finally, we make use of a simple reset mechanism to allow autonomous learning: At the end of each episode, if the robot has fallen (as inferred from the IMU orientation estimate), we use a robust pre-programmed feed-forward controller to stand up. On rare occasions multiple attempts may be required. The robot does not need to be moved and simply continues from the location where it ended the previous episode. See Section 2 in the supplementary material for a detailed overview.

### 3.2  Robot Platform

We focus on the off-the-shelf OP3 bipedal robot platform [23], or its simulated analog modelled in MuJoCo [24]. This platform is built of 20 Dynamixel servo motors (XM430-W350-R) that we use in *position control* mode. All controllers use 20Hz control frequency. In real robot experiments, all controllers run directly on the onboard Intel NUC i3 computer using ROS [25] to send commands to the actuators. While the low weight and low cost of the robot are advantageous for real robot learning experiments, the low actuator and sensory specifications increase the difficulty of the learning task.

### 3.3  Distributed Data Efficient Reinforcement Learning

The input to our policy consists of proprioceptive observations listed in Table 1, and the two most recent RGB camera observations down-sampled to 64x64 resolution. The output of the policy is a distribution over 20 dimensional actions corresponding to joint position commands.

| Observation | Dimension |
|---|---|
| history of joint positions | $5 \times 20$ |
| history of IMU accelerations | $5 \times 3$ |
| history of IMU angular velocities | $5 \times 3$ |
| history of egocentric gravity directions | $5 \times 3$ |
| history of past actuator commands | $5 \times 20$ |
| history of linear velocity estimates (see [8]) | $5 \times 3$ |
| heading change over 500ms | 2 |

Table 1: List of proprioceptive observations received by our policies.

Our policies are trained using an asynchronous distributed version of the actor-critic MPO algorithm [1, 26] which uses a distributional critic [27] (DMPO), and which has shown to be effective in addressing high-dimensional tasks with continuous action space. The algorithm is implemented in JAX [28] using the code[2] released as part of [29] as a reference. All hyperparameters are listed in Section 5 in the supplementary material.

The robots' onboard computers run *actor* processes which synchronize the policy's neural network weights with a remote *learner* process at the beginning of each episode. After an episode of executing the policy on the robot, the actor process sends all data experienced in that episode to a remote replay buffer implemented using the Reverb library [30]. The learner process continuously reads data from the replay buffer, and updates the network weights via gradient descent on the DMPO actor and critic losses. In order to prevent overfitting, the learner process is blocked when the total number of gradient steps exceeds some threshold times the number of control steps executed by all robots. In our experiments, this threshold is $1/4$. The remote learner process runs in the cloud. Communication with the robots is implemented using the Launchpad API [31].

One advantage of this setup is that we can scale from one robot to multiple robots in a practical and application-relevant way. But it also shifts the computational load away from the robot and the location of the experiment. This in principle allows to learn with robots having weaker onboard computing power, basically requiring only an internet connection to allow learning experiments.

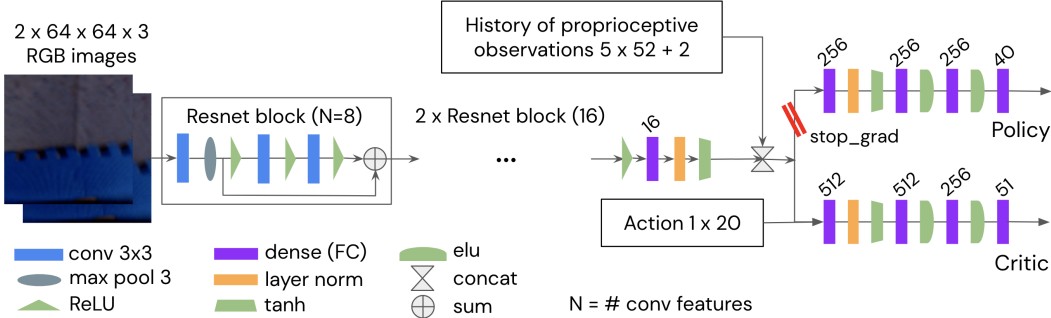

Figure 2: Network architecture. A 3-block Resnet produces image representations, which are concatenated with proprioceptive observations, and fed to separate policy and (action-conditioned) critic networks; see text for more details on tensor shapes and losses.

### 3.4 Network Architecture

The network architecture is depicted in Figure 2 and consists of an image encoder shared between the policy and critic networks. It processes the latest two RGB images received from the robot's camera. The outputs of the encoder are concatenated with proprioceptive observations (see Table 1) and processed by separate policy and critic networks. The shared encoder is trained using gradients from critic losses only. The output of the critic is 51-dimensional corresponding to the logits pa-

---

[2]https://github.com/deepmind/acme/tree/master/acme/agents/tf/dmpo

rameterizing the distribution of future returns [27]. The policy outputs the mean and the standard deviation of a diagonal Gaussian distribution from which we sample the actions for the actuators.

# 4 Experiments

For safety reasons and to ensure autonomous learning, the workspace in our experiments consists of a rectangular court delimited by foam walls. This prevents the robot from walking away, and, in combination with stand-up resets, represents a very effective way to minimize human intervention. In some sense, this is similar to the play pens used for babies when they start learning to walk. Except for this, all other environmental conditions are uncontrolled, e.g. we train in different daylight conditions, with various furniture items present around the workspaces, and different court sizes (see Figure 1). To avoid entanglement with the power cable, real robots are powered by a battery pack which needs to be changed roughly every 30 min.

At the beginning of training, the policy explores randomly which causes the robots to fall very often. To reduce the risk of damaging the hardware, we equip the robots with a 3D-printed body armour covering the front and back of the torso. In addition, we limit the range of the joints to prevent self-collisions that might damage individual actuators or other hardware parts. We also apply an exponential filter on the actions sampled from the policy (see Section 6 in the supplementary material). History of these filtered actions is included in the policy's inputs (see Table 1) in order to mitigate partial observability due to delays.

## 4.1 Simulation Experiments

A simulation environment is used to make preliminary tests ahead of the real world experiments. Aside from using it to tune network architecture and hyperparameters, we want to know if the sample complexity is within a reasonable range and whether vision can be used as viable alternative to instrumented state estimation.

We roughly replicate the real world setup – a $4 \times 4$ m court surrounded by walls – in simulation (see Figure 3). We randomize the textures for the floor and walls in every episode to roughly mimic the diversity of visual inputs in the real world setup where the learner receives trajectories from actors running in different locations and under various lighting conditions. The position and orientation of the robot are randomised at the beginning of every episode. We do not make additional efforts to make this simulated environment more realistic as we do not aim for sim-to-real transfer (see Section 1). Given the lack of rich textures, realistic lighting conditions, and motion blur it would be unlikely for a policy to transfer (see Figure 6 in supplementary material).

Figure 4 depicts the training curves of three different agents: 1) a *vision* agent using raw images, 2) a *pose* agent having access to ground-truth robot pose (position and orientation), and 3) a *blind* agent without additional non-proprioceptive measurements. All are trained in an open square court but tested either with or without an additional disturbing wall (see Figure 3). Starting with the *blind* agent, we can observe that the attained performance is only marginally affected by the presence of an additional wall because it learns a behavior that is robust to changes of wall configuration.

With access to ground-truth pose (*pose* agent), the robot learns very quickly to walk within the court. But when adding a wall at test time the performance strongly degrades as the robot collides with the wall and is unable to recover. This is improved by the proposed *vision* agent. While performing almost on par with the *pose* agent in the simple square court, it is also able to naturally generalise when the wall configuration is altered. This shows that there is a significant benefit when including vision in real world settings were space is usually limited and more cluttered. The difference in behavior can also be appreciated when visualising the policy (see Figure 3): The policy with vision avoids the wall (which was not present during training) and turns into a different direction (see Section 7 in the supplementary material for further details). Overall, the *vision* agent achieves good performance after about 1M environment steps. On a real robot, and taking into account repair and maintenance, this could be collected within 5 days.

Given the limited space of our environments all experiments have the tendency to develop circular motion patterns (see Figure 3) despite the "no turning" component of our reward. This is expected– we only use the "no turning" reward to discourage walking in very small circles which would prevent the robot from exploring the workspace and interacting with the walls.

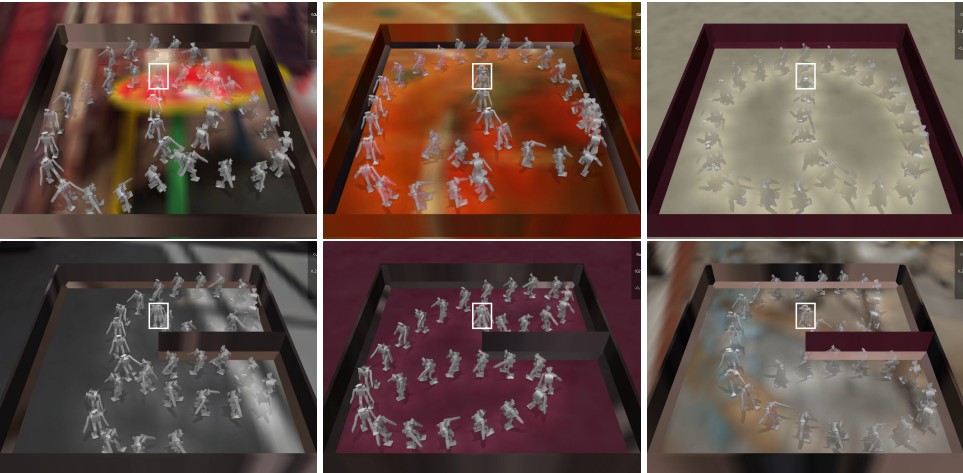

Figure 3: Overview of the simulation setup. Our simulated model of the OP3 robot is in a $4 \times 4$ m square court with random textures. Top: training setup. Bottom: test setup with additional wall to observe the reaction of the robot. We also visualise trajectories of the best performing *vision* agent which is also the one we train on the real robot. One can observe the circular motion that is learned and how the robot avoids walking into the walls. The white square highlights the start position.

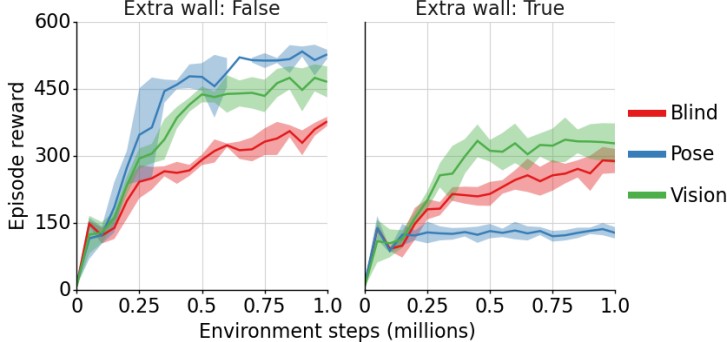

Figure 4: Simulation results comparing agents trained with vision and proprioceptive sensing (green), with groundtruth pose and proprioceptive sensing (blue), and with proprioceptive sensing only (red). The right plot shows the performance of the same agents in a court with an additional wall which was not present during training. The shaded area corresponds to 95% confidence interval across seeds.

## 4.2 Real robot experiments

Our robot experiments focus on sample complexity relative to simulation experiments, on assessing whether our trained policies make use of vision, and on evaluating the degree to which they can generalize to different robots and environments.

Figure 5 (left) shows the learning curve for the distributed two-robot experiment. We plot the episode results experienced by different robots in different colors to demonstrate that the robots had to be occasionally stopped for repairs (while the other robot would keep collecting data). Interesting behaviors start to emerge after about 25k environment steps of training which corresponds to 2-3 hours of experiment time. This includes battery changes, time spent on resets and logging data in between episodes. It takes much longer to collect 25k environment steps at the beginning of the experiment than at later stages because the robot falls a lot and the experiment time is dominated by the "stand up" reset behaviors (see Figure 5 in the supplementary material). At this point the robot has learned to balance and to avoid falling which causes the episode lengths and the returns from the "upright" component of the reward to increase. From this point on, the robot starts taking

first steps, steadily improving its walking speed and stability. It develops a bias for turning right and converges to circular walking behavior inside the enclosed workspace. See supplementary video for recordings from the experiment.

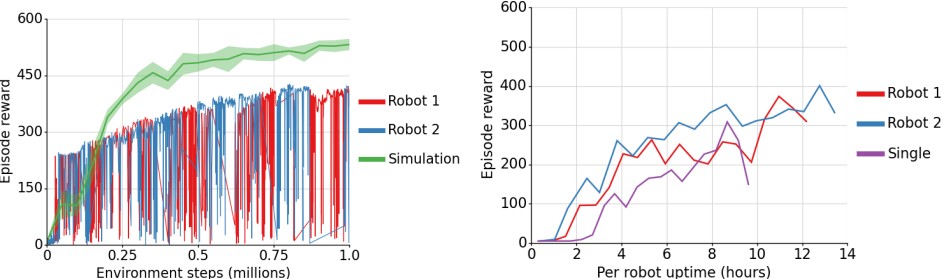

Figure 5: **Left**: Training curve for a distributed experiment with two robots called *Robot 1* and *Robot 2*. The episode rewards achieved by each robot are plotted in different colours. We do not smooth the curves to preserve information. The intervals where only one colour appears correspond to downtime of one robot. We also show a learning curve from an analogous simulation experiment. **Right**: Episode reward plotted against per robot uptime, comparing a 2-robot experiment and an independent single robot experiment.

Figure 5 (left) also shows a learning curve from an analogous simulation experiment. Once the robot learns to avoid falling, the rate of learning progress is about half that of the simulation. We believe this gap is caused by several factors which make the real environment more complex than the simulation: inaccurate model of delays in simulation; changing robot dynamics due to wear and tear of the joints causing them to become loose and eventually completely broken; changing robot dynamics due to battery voltage fluctuations; and non-stationary lighting conditions.

We also ran a shorter, single robot experiment and observed a similar sample complexity as that of the two-robot experiment (see Figure 4 in supplementary material). This is not obvious a priori because larger diversity of inputs (as is the case in the two-robot experiment) typically renders reinforcement learning more difficult. The single robot experiment also serves as an additional "seed" and provides evidence for the reproducibility of our setup. To demonstrate the increased throughput of the two-robot experiment, Figure 5 (right) shows the dependence of individual robots' performances on their uptime, and compares it to the same curve for the robot in the single robot experiment. We see that individual robots in the two-robot experiment, where data is shared, improve faster, pointing to a promising direction of scaling up real experiments.

**Visual wall avoidance** To demonstrate that the learned policy uses camera observations to avoid walls, Figure 6 depicts the reaction of the robot when encountering the wall. Because of the policy's right turning bias, we initialize the robot so that a wall is on its right (red square marks the robot's initial position). If no wall is present, then the robot turns in a large circle (top row). In comparison, when a wall is present, the robot takes a sharper turn as to only just brush the surface of the wall (bottom row).

**Further robustness tests** We evaluated the final policy obtained in the two-robot experiment in two additional settings: 1. on a third robot which was not used during training, and 2. on the same robot which was used in the experiment but using a black floor instead of a blue one. Table 2 shows the results of this evaluation. The policy works well on *Robot 3* achieving only a slightly lower per step reward than the robot used during training (*Robot 1*). However, it is less stable on *Robot 3* causing the robot to fall more frequently. While this difference can be due to transfer, it is likely that the state of *Robot 3* was unfavorable – as mentioned before, the dynamics of the robots vary across time and things like loose joints or screws can impact what a robot can do.

The policy performs considerably worse on a black floor demonstrating that the visual module is adapted to the blue floors encountered during training (also see supplementary video). However, as demonstrated in the simulation experiments, this could be improved by exposing the agent to black floors during training as well.

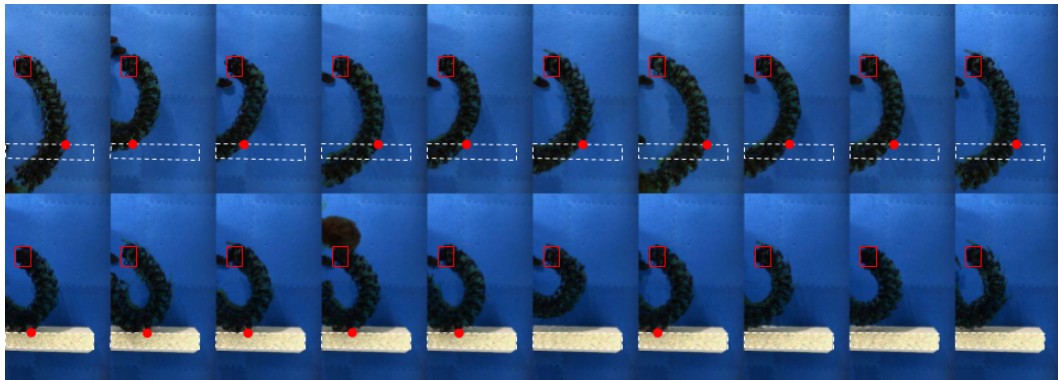

Figure 6: Qualitative comparison of robot trajectories when encountering a wall. The robot starts in the red square facing right. It then starts walking while slightly turning towards its right. The robot makes a sharper turn when there is a wall (beige foam block, bottom row) than when we remove the wall (the white dots show where it would be). The first point of contact with the (potentially virtual) wall is depicted by a red dot. In the presence of an actual wall contact can either be avoided or is shifted towards the left as to only just brush the wall.

| Evaluation context | Per step reward | Episode length |
|---|---|---|
| Robot 1 | $0.408 \pm 0.002$ | $857 \pm 151$ |
| Robot 3 | $0.389 \pm 0.003$ | $520 \pm 160$ |
| Black floor | $0.318 \pm 0.005$ | $136 \pm 36$ |

Table 2: Evaluation of the learned policy under various contexts.

## 5   Conclusions

In this paper we showed that we can learn a walking gait for a bipedal, off-the-shelf robot, directly on hardware, outside a well-controlled lab infrastructure. Using a state-of-the-art RL method (DMPO) renders learning with minimal prior knowledge feasible in terms of data efficiency.

Scalable reinforcement learning in the real world is still in its infancy, and to the best of our knowledge, this is one of the largest scale experiments for model-free learning for legged robots. Limiting ourselves to on-board sensors only and collecting data from multiple robots at various geographic locations allows us to scale up the diversity of environments which the robots encounter. While in this paper we used only two locations with different lighting conditions, arena sizes, and background visuals, in future this can include many more locations with varying terrains and obstacles. This should be contrasted to lab settings with immobile equipment for external state estimation. We further also intend to remove the requirement for a reset controller by learning to stand-up.

Specifically, we plan to explore learning more dynamic behaviours, as well as tasks specified by rewards derived from camera images such as searching for an object in a cluttered environment. Such tasks require memory, complex visual processing, and longer horizon decision making, hence it will be necessary to train for much longer than 1M steps. We believe that the presented approach is well suited for learning at such scale. We also plan to follow a "curriculum" and progressively move, or completely remove, the protective walls.

**Social Impact Statement**   Embodied agents using perception and locomotion have the potential to enable applications with significant positive social impact. E.g. they can be deployed in search-and-rescue operations (required in force majeure events like earthquakes, landslides, nuclear disasters), infrastructure inspections that might pose risks for humans (e.g. assessing if a bridge or tunnel is in a healthy condition), and many others. However, there is also the risk that AI-driven technology in general can be used for harmful purposes, e.g. in military applications. We strongly oppose such use cases. We believe that our particular application presents a low risk of being used in such situations as it has limited potential for causing large-scale damage, but we actively support discussions around safety and fairness in Robotics.

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
