# OpenReview forum: "Towards Real Robot Learning in the Wild: A Case Study in Bipedal Locomotion"
_robot-learning.org/CoRL/2021/Conference — CoRL2021 Poster_

### Official Review · Reviewer_tDwM · 2021-07-17

**Originality:** Good
**Technical Quality:** Very Good
**Clarity Of Presentation:** Very Good
**Impact:** 3

**Recommendation:**

Weak Accept: I recommend accepting the paper, but will not argue for my recommendation if the majority of other reviewers have a different opinion.

**Summary:**

This paper presents a system to train bipedal robot policies directly on hardware, with only on robot sensors, including an RGB camera. Result demonstrates that it is possible to train bipedal policies from scratch on hardware, including learning behaviors such as wall avoidance.

**Issues:**

1. It will be interesting to know how much hyperparameter tuning is necessary in simulation before deploying the system on the real robot. e.g, the reward weights, the history length of the observation. It is not clear to me that I can arbitrary choose these parameters (within reasonble ranges) and the system will still work well on the real robot (or simulation). The supplementary material provides ablation on the resnet module, which seems to indicate tuning in simulation is a necessary first step.


**Reviewer Expertise:**

Very good: Comprehensive knowledge of the area

**Strengths And Weaknesses:**

Pro:

1. Learning bipedal locomotion policies on hardware, with RGB camera as sensors.

Con:

1. A manually defined reset controller is still needed. To truly "learn from scratch", it feels like this reset controller should also be learned. And it is not clear how this reset controller should be constructed for larger legged robot. For example, generating a stand up controller for a large quadruped is not trivial (e.h, Learning agile and dynamic motor skills for legged robots, by Hwangbo et al), not to mention a large biped.

2. It feels like using simulation to tune some of the hyper-parameters is still necessary. (Please correct me if I am wrong).

**Summary Of Recommendation:**

While this paper presents an approach to learn a policy directly on real bipedal robots (and maybe the first one to do so), one could raise the question whether "learning from scratch" is actually true.

First, a reset mechanism (in this case, a manually constructed controller)  is still needed. And given the complexity of the policy structure, it seems like a simulation is still needed to tune hyperparameters before deploying it on the robot, so the claim that "largely lift the assumption of the availability of a simulation model" is not completely true. This raises the question, since we already need a simulation anyway, why not warm start from a policy trained in simulation instead of wasting additional efforts by training from scratch?

That being said, it is still interesting to see a policy can be learned on hardware directly within reasonable amount of time. Allow training on hardware is definitely valuable (wether it is training from scratch or warm starting), and this paper presents a system to do that.  And using a simple RGB camera is enough for learning wall avoidance is also a nice contribution.

---

> ### Author Response · Authors · 2021-08-26
> **Response to reviewer tDwM**
>
> We thank the reviewer for their time and their critical and constructive comments.
>
> **Requirement for resets**
>
> _The reviewer questions how relying on a manual stand-up controller affects the applicability of the method._
>
> We fully agree that the stand up behaviour should be learned and this is and will be part of both ongoing and future work. Robot resets do indeed represent a significant challenge for real world robotics RL. In the present work we approach the issue by using an onboard stand-up controller. Given that a significant percentage of robots come with such a stand-up controller, we believe that this is a reasonable temporary simplification. In particular, the present work shows how to address challenges around observations, reward computation, and data efficiency and confirms this with real world experiments. We have added a comment on this matter to the paper.
>
> **Hyperparameter tuning in simulation**
>
> _The reviewer raises concerns around the requirement of a simulator for hyperparameter tuning._
>
> It is true that a simulated environment is useful for hyperparameter tuning. However, our experience with DMPO suggests that the default hyperparameters work well across many environments (see e.g. [Hoffman 2020]) and only a small-scale hyperparameter tuning is needed in new environments. Thus, tuning only requires an easy-to-obtain simulator which implements the reward function but does not need to accurately model physics. The same applies to tuning the vision architecture: It is conceivable to tune the vision architecture in a completely different environment which does not even contain the robot -- it just needs to capture the expected diversity of visual inputs. This is emphasized by Fig. 3, for instance, which shows the simulation environments that we use for tuning the vision architecture and which are visually very different from the environment in which the robot is eventually deployed in the real world.
>
> **Warm starts**
>
> _The reviewer asks whether warm starts could be used._
>
> We emphasise that we use an off-the-shelf rigid body dynamics simulator to merely develop a reward function, estimate sample efficiency, and tune the learning algorithm. We believe that the knowledge gained regarding these points transfers to other robots of similar morphologies, and thus the requirement of a simulation model can be lifted when applying our method to other bipedal robots. We also verified that even blind policies trained with a heuristic domain randomisation in sim did not transfer well to the real robot, and we concluded that we would need to invest considerable amount of time into improving the simulation (sophisticated graphics, motion blur, better contact and delay models) as well as into tuning the domain randomisation strategy to make progress with sim2real. Ultimately, we think that sim2real transfer could be useful for pretraining but due to the complexity of the real world we doubt that reinforcement learning directly on the hardware can ever be taken out of the equation, especially in more complex environments. Therefore, it is important to understand how to implement learning directly on the hardware and that is what we chose to study in this paper.
>
> **Summary**
>
> We hope we have clarified the reviewer's concerns regarding resets and the role of simulation. Please let us know if there are any points you would like us to expand on further.

---

> > ### Comment · Reviewer_tDwM · 2021-08-26
> > **response to rebuttal**
> >
> > "Given that a significant percentage of robots come with such a stand-up controller, we believe that this is a reasonable temporary simplification"
> >
> > By that logic, a significant percentage of robots come with a walking controller as well, why does this paper bother to learn walking at all? To be clear, I am not against learning walking, as indicated by my initial rating of this paper, my argument is just following the logic of the authors' rebuttal. And the authors' rebuttal doesn't address my concern that this paper is not learning a controller "from scratch", as claimed in the paper.
> >
> > "tuning"
> >
> > A lot of things need to be tuned here. I believe the claim that hyperparameters for the learning algorithms, architecture need minimal tuning, but what about the reward function? I have no problem with the claims that no accurate simulation model is needed, but still your simulation needs to be approximately close to your robot to tune your reward function, etc. Unless the authors are able to show that keeping everthing the same, this framework can work for other robots, even in simulation.
> >
> > "it is important to understand how to implement learning directly on the hardware and that is what we chose to study in this paper"
> >
> > I am not disputing the merit of implementing learning directly on the hardware, as indicated by my original rating. I am just disputing the merit of learning from scratch on hardware (with emphasis on "scratch" here). Currently the paper is neither completely "learning from scratch" nor utilizing some form of warm starting. To show benefits of learning from scratch, I will suggest using warm starting and show that warm starting actually lower the final performance on hardware (similar to alphazero is better than alphago because it doesn't wam start with imitating human players).

---

> > > ### Author Response · Authors · 2021-08-27
> > > **Response to comments**
> > >
> > > Thank you for your fast reply. Please find below our response to your three comments.
> > >
> > > ### Re. learning from scratch:
> > > Thank you for pointing this out. We agree that the term “from scratch” is not well defined. We removed it from the paper and now we only refer to our method as learning directly on hardware. Our use of “from scratch” is consistent with prior work where it was used to describe model-free reinforcement learning on hardware even if it required scripted resets or other prior knowledge (see e.g. [1, 2]), but you are right that it is better to avoid such ambiguous wording. We hope that this addresses your concerns.
> > >
> > > [1] Haarnoja et al., 2018, Learning to walk via deep reinforcement learning
> > >
> > > [2] Riedmiller et al., 2018, Learning by playing-solving sparse reward tasks from scratch
> > >
> > > ### Re. tuning and simulation:
> > > The walking component of the reward is based on kinematics and has no free parameters. It was applied to other robot morphologies in Hafner et al. 2020. The two reward hyperparameters are the relative weights of the upright and “don’t turn” rewards, and we agree that simulation is needed for tuning these. However, we stress that essentially any simulation which respects rigid body dynamics and has an approximate contact model is sufficient (the contact dynamics don’t need to match the real system, e.g. we made no attempt to match the friction in simulation). Such approximate rigid body simulators are widely available so we believe it is reasonable to use them for visualization and basic tuning.
> > >
> > > ### Re. warm-starting:
> > > We apologize if our previous response has caused confusion. We did not mean to dispute the value of warmstarting.
> > >
> > > Although we do think that minimizing the prior knowledge that a learning system requires is a goal that we should strive for, we do not claim that our current approach achieves this goal, nor do we claim that it is the right objective in every situation. Indeed, warmstarting from a simulation is a direction that we are actively investigating.
> > >
> > > The goal of the present paper is to study how to achieve efficient learning directly on hardware with as little instrumentation of the environment as possible. This question is orthogonal to the question of whether and how learning on the hardware can be accelerated via warm-starting. Considering that warmstarting comes with its own algorithmic complications we believe that it makes methodologically sense to investigate this question separately using an algorithm with fewer knobs. Hence the focus of the present paper.

---

> ### Comment · Reviewer_tDwM · 2021-08-31
> **keep score unchange**
>
> The authors' response addresses most of my concerns and I am keeping my (positive) rating unchanged.

---

### Official Review · Reviewer_n7gp · 2021-07-24

**Originality:** Good
**Technical Quality:** Good
**Clarity Of Presentation:** Very Good
**Impact:** 3

**Recommendation:**

Weak Accept: I recommend accepting the paper, but will not argue for my recommendation if the majority of other reviewers have a different opinion.

**Summary:**

The paper presents a case study of learning locomotion skills on a small table-top size robot, Darwin OP3. The key achievements include 1) deploying DMPO, a data-efficient off-policy deep RL algorithm, on real robots, 2) training a policy from two robots in different geographic locations, and 3) training a visual policy out of real-world experience. As far as I know, these have not been achieved for real-world robot learning. The authors have demonstrated that the proposed on-robot learning system can train an effective locomotion policy on a small humanoid robot.

**Issues:**

I made most of the suggestions above: please refer to the comments.

**Reviewer Expertise:**

Excellent: Expert knowledge on the topic of the paper

**Strengths And Weaknesses:**

+ The paper demonstrates a practical case study of learning bipedal locomotion from real-world experience.
+ The paper achieves some state-of-the-art results, such as learning bipedal locomotion, multi-robot learning, or learning visual policies from real-world experience.
- The techniques in the paper are not too novel (because this is a case study).


**Summary Of Recommendation:**

This is a case study paper that investigates the scenario of deploying deep RL algorithms to on-robot learning environments. In summary, the presented experiments are quite interesting and can serve as a good data point for other researchers. However, we have to admit that the paper does not contain much technical novelty.

The most impressive result is visual navigation. Although it is hard to say that the robot learns a precise visual navigation skill: rather, the robot seems to estimate the distance to the goal from RGB images and tries to avoid the collision. But I can imagine that visual inputs would also suffer from the sim-to-real gap. It will be great if the authors can highlight the sim-to-real gap a little bit more by comparing simulation-generated images and real observations. It might be pretty obvious, but better to be explicit about it.

Another interesting contribution is to learn a locomotion skill based on proprioceptive observation. I am a little bit surprised (in both good and bad ways) that it works for real bipedal locomotion. It might be a little bit better if the authors can discuss this issue more. Why can we assume non-slip conditions? I guess locomotion of many small bipedal robots accompanies foot slips. Would it be possible to measure the accuracy of this reward function design by comparing it against the groundtruth mocap data?

I also want to see more discussion about the resetting. Human interventions are also needed when the robot is out of bound (happens very often when the policy is matured) or stuck at the corner. Did you also conduct manual resets in this case? I would suggest the authors describe their approaches.

Simulation results are not presented in the supplemental video. It might be good to include a couple of simulation results as well.

---

> ### Author Response · Authors · 2021-08-26
> **Response to reviewer n7gp**
>
> We thank the reviewer for their interest in our work and their insightful feedback.
>
> **Case study or theoretical advance?**
>
> We agree with the reviewer that this is indeed a case study. As such the proposed work brings together different state-of-the-art methods in order to tackle the problem of learning bipedal locomotion directly on real hardware. In particular we show that it is possible to learn vision-guided locomotion, from scratch, with minimal instrumentation and intervention, something that has not been achieved before. We believe this is an important demonstration since it pushes the boundaries of what was thought possible with deep RL, and also outlines an angle of attack on the problem of learning dynamic robot behaviour in the wild.
>
> **Sim-to-real gap**
>
> _The reviewer asks us to highlight the visual sim-to-real gap._
>
> This is indeed an important point, thank you for pointing this out. As also explained in the response to reviewer tDwM we only use simulation for ablations and to coarsely optimize some hyperparameters of our architecture. We do not make an attempt e.g. to match the visual appearance of the simulation environment to the real world. The resulting visual sim2real gap should be clear by comparing the Fig. 3 (simulation environment) with Fig. 1 (real world setup) but is made worse by differences in the camera resolution, varying lighting conditions and motion blur. We will emphasize this more in the manuscript, by including a side-by-side comparison of first person observations of the robot in simulation and the real world.
>
> **No-slip assumption**
>
> _The reviewer is surprised that the no-slip assumption can be employed._
>
> This issue is discussed in [Hafner 2020], where their main finding is that even when slippage is present, the signal that the reward provides is strong enough to motivate forward locomotion. We can confirm this finding in the present work. Our experience is also that while slippage causes noisy reward, the agent will still achieve forward locomotion when averaging across the collected experience.
>
> **Resets**
>
> _The reviewer asks for more details on the effectiveness of the resets._
>
> The combination of (1) foam walls and (2) the stand-up controller was very effective in minimizing human intervention. We never had to manually move the robot as the foam walls would prevent it from leaving the workspace and the stand up controller provided by the vendor was always able to initialize the robot to an upright pose. In case a reset would fail, the robot would just try again and complete the stand up reset after at most 2 or 3 tries. Regarding collisions with walls, this would often make the robot fall, which would naturally lead to a randomisation of the initial position for the next episode. Also, as the agent matures, it learns to stay away from the walls. We have added comments on these points to the paper.
>
> **Simulation videos**
>
> _The reviewer asks whether we can include simulation videos._
>
> We have added videos showing the simulation results. These correspond to the trajectories visualized in Figure 3 of the supplementary material.
>
> **Summary**
>
> We hope to have answered the reviewer's concern and thank them again for the helpful review.

---

### Official Review · Reviewer_Mhfu · 2021-07-24

**Originality:** Fair
**Technical Quality:** Poor
**Clarity Of Presentation:** Fair
**Impact:** 2

**Recommendation:**

Weak Reject: I recommend rejecting the paper, but will not argue for my recommendation if the majority of other reviewers have a different opinion.

**Summary:**

The paper describes the framework to produce walking gaits from on board proprioceptive sensors of an off-the-shelf small humanoid robot, by using Distributional Maximum a posteriori Policy Optimization (DMPO) and processing of visual data directly to policy generation.

The paper highlights that the reinforcement learning training is executed on the robot in real time with visual input data processed through frames with 3-block Resnet networks for the policy and critic networks.

The experiments show that the robot learns to walk while avoiding collision with the walls, and can benefit from sharing the learning experience with another robot.


**Issues:**

More mathematical background is required for the formulation and justification of the proposed reinforcement learning method. The idea of using a vision system for the walking gait is relevant, as well as a discussion about the scalability of the method.


**Reviewer Expertise:**

Good: General knowledge of the area

**Strengths And Weaknesses:**

Strengths:
The inclusion of proprioceptive sensing provides another venue for the learning of gaits and navigation for humanoid robots.
The policy training was realized in real time, so the robot had to be provided with a standing up strategy that allowed it to keep training.
It is stated how the vision system improves on the ground truth pose when walls are added, providing evidence of the importance of the vision system.

Weaknesses:
The paper lacks a theoretical formulation to support the generality of their learning method. Could the experiments be scaled for more complex bipedal robots?
There is little detail about the weights of the reward function or the reinforcement learning action space.


**Summary Of Recommendation:**

The paper requires to be written with a proper formulation of the reinforcement learning algorithm that can be generalized to more complex or different bipedal systems. Also, there are many missing details on the construction and training of the neural networks and their action on the robot.

---

> ### Author Response · Authors · 2021-08-26
> **Response to reviewer Mhfu**
>
> We thank the reviewer for their time and insightful feedback.
>
> **Theoretical background**
>
> _The reviewer asks for more theoretical background regarding the RL agent and how this supports the generality of its application._
>
> As detailed in section 3.3 of the paper, the agent is DMPO [Hoffman 2020] and theoretical details about the agent are available in references [Abdolmaleki 2018] and [Bellemare 2017]. A link to an open-source code for DMPO is on page 4 of the paper. DMPO is a state-of-the-art RL agent that has proven to be applicable to a wide array of continuous state-action space problems and high dimensional action spaces in simulation tasks [Hoffman 2020].
>
> We have further emphasized these points in the paper and will add a short overview of the DMPO framework to the appendix.
>
> We would also like to highlight that the contribution of the present paper is not the algorithm per se, but rather that DMPO can successfully solve a continuous and high-dimensional *real* robotic task, in a distributed and autonomous learning environment.
>
> **Complexity of the robot**
>
> _The reviewer asks whether the algorithm would be applicable to more complex legged robots._
>
> Although the robot that we consider in this paper is small in stature, it already has considerable complexity. The robot has 20 degrees of freedom (c.f. 12 DoF for ANYbotics’ ANYmal and 28 DoF for Boston Dynamics’ Atlas). In addition, its actuators and sensors are comparatively low specification and have large delays, which adds further complexity to the control problem. Therefore we believe that our algorithm does scale to other highly complex robots. We thank the reviewer for raising this question, and have added short commentary on this matter to the manuscript.
>
> **Details on the network, training, and rewards**
>
> _The reviewer asks for details on the construction and training of the neural networks._
>
> Figure 2 in the paper provides all network architecture details, and we list all training hyperparameters in Table 2 of the supplementary material. A derivation of the reward is available in Section 4 of the supplementary material together with the weighting of the different reward terms.
>
> **Summary**
>
> We thank the reviewer for their review and hope that we have been able to answer their questions. We would kindly ask them to let us know if any further details are required in order for them to consider increasing their rating.

---

### Meta-Review · Area_Chair_oShQ · 2021-08-12

**Recommendation:** Accept (Poster)
**Confidence:** 3

**Metareview:**

The paper is borderline. Following the authors' responses, two of the three reviewers are positive. The AC is also positive, in part due to the focus on experiments with real hardware, which is encouraged at CoRL. The AC supports the majority recommendation.

---

> ### Author Response · Authors · 2021-08-26
> **Response to all reviewers and area chair**
>
> In this paper we have presented a first demonstration of deep learning for perception-guided bipedal locomotion directly on real hardware. In particular, we have provided solutions to several important practical concerns in this setting: e.g. how to learn when only relying on the robot’s on-board sensors, how to not require external state estimation or other instrumentation of the environment, and how to minimize the need for human intervention.
>
> We are glad to see that the reviewers found the proposed approach interesting and relevant. The reviewers have appreciated the scope of the problem and the results but would like further details regarding the learning framework, resets, and more discussion on the limitations of the setup. Two of the three reviewers also found the presentation very good and clear.
>
> As a general comment we would like to emphasise that we believe that the success of robot learning in the real world is dependent on a combination of (a) efficient algorithms and (b) the ability to address questions related to resets, reward definition, state estimation and instrumentation of the learning setup to enable learning with minimal human effort. In this paper we have specifically focused on these latter questions.
>
> We further believe that it will be hugely impactful if we can enable robot learning, in the real world, without human effort. It will enable flexible adaptation of robot behaviour during deployment, and reduce, or even eventually obliviate, the need to produce highly accurate simulations. To understand and push the boundaries in this direction we have therefore deliberately taken an extreme stance in that we focus on end-to-end learning on the hardware only, and do not utilise warm starts.
>
> Below we provide point by point responses to the reviewers’ queries.

---

### Decision · Program_Chairs · 2021-09-13

**Decision:**

Accept (Poster)

**Comment:**

The paper is borderline. Following the authors' responses, two of the three reviewers are positive. The AC is also positive, in part due to the focus on experiments with real hardware, which is encouraged at CoRL. The AC supports the majority recommendation.